# The rising complexity and burden of multimorbidity in a middle-income country

Shamini Prathapan[1], Gunasekara Vidana Mestrige Chamath Fernando[2,3]*, Anne Thushara Matthias[4], Yashodara Bentota Mallawa Arachchige Charuni[4], Herath Mudiyanselage Gayan Abeygunawardhana[3], Batheegama Gamarachchige Gayasha Kavindi Somathilake[2]

1 Department of Community Medicine, Faculty of Medical Sciences, University of Sri Jayewardenepura, Nugegoda, Sri Lanka, 2 National Centre for Primary Care and Allergy Research, University of Sri Jayewardenepura, Nugegoda, Sri Lanka, 3 Department of Family Medicine, Faculty of Medical Sciences, University of Sri Jayewardenepura, Nugegoda, Sri Lanka, 4 Department of Medicine, Faculty of Medical Sciences, University of Sri Jayewardenepura, Nugegoda, Sri Lanka

* chemetf@sjp.ac.lk, chemetf@gmail.com

## Abstract

### Background

The limited knowledge on aetiology, epidemiology and risk factors for multimorbidity especially evident from low and middle-income countries curtail the development and implementation of sustainable healthcare models. Sri Lanka, boasting for one of South Asia's most efficient public health systems that is accessible free-of-charge by the citizens is presently transitioning from lower-middle to upper-middle-income tier. Faced with the triple burden of disease, it is imperative for Sri Lanka to incorporate an integrated model to manage multimorbidity.

### Methods

A descriptive cross-sectional study was carried out in medical clinics of a tertiary care hospital and a University primary care department. Data were extracted on to a form from the clinical records of patients over the age of 20 years with at least one non-communicable disease (NCD) and analysed.

### Results

Multimorbidity was present among 64.1% of patients (n = 1600). Nearly 44.44% of the patients aged 20–35 years have a minimum of two disorders, and by the time they reach 50 years, nearly 64% of the patients have two or more non-communicable diseases. Nearly 7% of those aged over 65 years were diagnosed with four or more disorders. A fourth of the sample was affected by co-morbid diabetes mellitus and hypertension, whereas the combinations of coronary heart disease with hypertension and diabetes mellitus were also found to be significantly prevalent. A salient revelation of the binomial logistic regression analysis was that the number of disorders was positively correlated to the presence of mental disorders 7.25 (95% CI = 5.82–8.68).

**Data Availability Statement:** All relevant data are within the paper and its Supporting Information files.

**Funding:** The authors received no specific funding for this work.

**Competing interests:** The authors have declared that no competing interests exist.

## Conclusion

Multimorbidity is highly prevalent among this population and seemingly has a detrimental effect on the psychological wellbeing of those affected. Therefore, the need for horizontal integration of all primary to tertiary care disciplines, including mental health, to manage multimorbidity by policymakers is emphasized as a priority task.

## Introduction

Management of the rising prevalence of chronic illnesses is one of the biggest challenges facing many countries worldwide. Individual diseases dominate healthcare delivery in many countries around the world and especially in Sri Lanka. People with multimorbidity—those with two or more chronic morbidities—require a more comprehensive approach [1].

Life expectancy has improved dramatically over recent decades. Between 2000 and 2016, global life-expectancy at birth, for both sexes combined increased from 66.5 to 72.0 years [2], and currently exceeds the age of 75 years in nearly 60 countries. However, the number of people with or at risk of long-term conditions, such as diabetes, mental health conditions, and cancer is also proliferating. People living with a chronic condition often have multiple rather than a single condition. As such, multimorbidity is common and has been rising in prevalence over recent years. In the UK, a large study revealed that more than 40% of the population (all ages included) had at least one long-term condition, and almost 25% of the entire population had more than one long-term condition [3].

Multimorbidity is becoming progressively more common with advancing age [4–6]. It is associated with high mortality, reduced functional status, and increased use of both inpatient and ambulatory health care [4]. The prevalence of multimorbidity in the world varies widely. In a systematic analysis of the prevalence of multimorbidity in high-income countries and low and middle-income countries, it was found that more than 50% of those older than 65 years had multimorbidity and that females were affected more [7].

Data on multimorbidity in South Asia is limited. With the increases seen in aging populations in Asian countries, South Asia is experiencing more multimorbidity than ever before [8, 9]. The prevalence of multimorbidity in South Asia varies from 4.5% to 83% [10]. The prevalence of multimorbidity in India, another South Asian country has been estimated to be 24% [9]. The only study done in Sri Lanka to date on multimorbidity has found a prevalence of 25.4% for cardiometabolic multimorbidity [11]. This has been conducted in rural Sri Lankan community setting. The prevalence of multimorbidity in an urban or a hospital setting in Sri Lanka has not been evaluated before.

Multimorbidity is a threat to patient safety [12]. Patients with multimorbidity are at a greater risk of safety issues for many reasons. Some of the reasons are polypharmacy, which may lead to poor medication adherence and adverse drug events, complex management regimens, more frequent and complex interactions with health care services leading to greater susceptibility to failures of care delivery and coordination, the need for clear communication and patient-centred care due to complex patient needs, demanding self-management regimens and competing priorities, more vulnerability to safety issues due to poor health, advanced age, cognitive impairment, limited health literacy and comorbidity of depression or anxiety. People have both physical and mental health issues simultaneously [13]. One systematic review that included 86 studies found that people with mixed mental and physical multimorbidity had the highest risk of active patient safety incidents and precursors of safety incidents [14].

The health care needs of patients with multimorbidity are complex. The successful management of them requires a shift away from specialism and more towards generalism. The association between sex, age and prevalence of specific chronic diseases is not established in Sri Lanka. A better understanding of the epidemiology of multimorbidity is necessary to develop interventions to prevent it, reduce its burden, and align healthcare services more closely with patients' needs. Assessing the multimorbidity will help put Sri Lanka onto the track of Universal Health Coverage. This study gives new information on the prevalence of multimorbidity in Sri Lanka. We aimed to examine the characteristics of individuals with multimorbidity (diagnosed with two or more NCDs) in terms of age, gender, socioeconomic dimensions and co-existing mental health disorders.

## Materials and methods

Written approvals to all the study procedures were sought from the Ethics Review Committee of the Faculty of Medical Sciences, University of Sri Jayewardenepura (ERC No:35/19). The investigators ensured that the study was conducted following the guidelines set out in the terms of reference and general management procedures of the said Ethics Review Committee, based on the International Guidelines on Biomedical Research of the World Health Organization (WHO) and the Council for the International Organizations of Medical Sciences (CIOMS). Consent was not obtained as there was no direct patient interaction and exclusively the anonymized data were extracted from the clinical records.

A descriptive cross-sectional study was carried out in the Colombo District of Sri Lanka. Sri Lanka is an Island located in the Indian Ocean, with a midyear population estimated to be with 22.235 million inhabitants. The allopathic system of healthcare in Sri Lanka comprises of a public and a private sector. The public sector services are available island-wide, whilst the private sector is based on market demand, and mostly concentrated in the urban areas of Sri Lanka. Free access to health care is a priority of the government of Sri Lanka, who has committed to maintaining this policy for the last two to three decades.

This study was carried out in the medical clinics of a tertiary care teaching hospital in Sri Lanka and a University primary care department (Family Practice Centre). These two study settings were selected as the patients in the suburbs of the University are cared in a coordinated manner between these two institutions through a referral and back-referral system, where a secondary level hospital rarely has any involvement. The tertiary level teaching hospital is managed by the central Ministry of Health, whereas the University managed the primary care department. Both these University-operated institutions are located in the southern part of Colombo, the commercial capital of Sri Lanka.

Data extraction was limited to the clinical records of adult patients (18 years or older) with a minimum of one non-communicable disease (NCD) diagnosed by either a consultant physician or a consultant family physician, and the most recent encounter occurred during the year 2019. Clinic records lacking any one of the following information; i.e. the age, the sex, area, drugs administered were excluded.

The study population was divided into four age groups; 18–35 years, 36–50 years, 51–65 years, 66 and more years. Since many NCDs were considered for multimorbidity, a prevalence (p) of 50% was used to obtain the largest sample size at 95% confidence level with 5% margin of error (e) using the equation $n = Z^2 p q/e^2$. A sample size of 384 was obtained for one age group of adults in order to extract data from a finite number of records while also yielding a sufficient statistical power. Therefore, a sample size of 1600 was obtained from both settings, 800 records from each setting with including all four age groups.

All clinic records from the 1st of January 2019 were scrutinized until the sample size was achieved. Investigators collected data from the clinical records of the two settings. Personally Identifiable Information (PII) pertaining to the patient, such as name or address was not extracted, and the anonymized records of each patient were assigned an alphanumeric identifier and kept in the safe custody of the investigators. A data extraction form (Annexure 1) was used to extract the data from clinic records.

Frequencies, percentages with 95% Confidence intervals (95% CI) and cross-tabulations were obtained using the R software tool whereas both R (R version 3.3.3 (2017-03-06) RStudio Version 1.3.959 © 2009–2020 RStudio, PBC "Middlemist Red" (3a09be39, 2020-05-18) for Windows Mozilla/5.0 (Windows NT 10.0; Win64; x64) AppleWebKit/537.36 (KHTML, like Gecko) QtWebEngine/5.12.6 Chrome/69.0.3497.128 Safari/537.36)and SPSS (IBM SPSS Statistics Version 20 IBM Corp. (2017). *IBM SPSS Statistics for Windows*. Armonk, NY: IBM Corp. Retrieved from https://hadoop.apache.org) software were used for the graphical display for descriptive analysis. Binary logistic regression was carried out after confirming that the data satisfied the relevant assumptions to examine associations between mental health and multimorbidity, restricting the analysis to those aged 20 years and older because mental health morbidities in children are rare. Adjusted odds ratios (ORs) and 95% CIs were used to report the analysis with the help of SPSS and R software.

## Results

The total sample size was 1600, of which approximately half were women (54%). Exactly equal proportions of patient records were included from primary care ($n_1$ = 800) and tertiary care ($n_2$ = 800) settings. The predominant age group was 51–65 years that accounted for 44% (n = 704), closely followed by the older persons over the age of 65 years (38%) as represented in Table 1.

Among the patients, 52.4% (n = 838) had diabetes mellitus, followed by 46.9% (n = 750) with hypertension. Table 2 illustrates the numbers and percentages of patients affected by individual NCDs.

Multimorbidity was present among 1026 (64.13%) of this group of patients. None younger than 20 years were found among the collected records. Nevertheless, it was an unfortunate yet a salient finding that by the time the population reached 20–35 years, 44.44% (24/54) of the patients have a minimum of two disorders and by the time they reached 50 years nearly 64% (178/280) of the patients have two or more non-communicable diseases. It was also evident in Fig 1 that by the time the population reached the age of 65 years, nearly 7% (41/614) have four or more disorders.

A significantly higher proportion of women (i.e. 5% more, Pearson Chi-Square = 6.97, p = 0.031) was affected by multimorbidity as compared to men (statistically significant

**Table 1. Socio-Demographical details among the group of patients.**

| Variable | Categories | Number of patients (n = 1600) (%) |
|----------|-----------|------------------------------------|
| **Gender** | Female | 863 (54%) |
| | Male | 736 (46%) |
| **Age** | 20–35 Years | 54 (3%) |
| | 36–50 Years | 226 (14%) |
| | 51–65 Years | 706 (44%) |
| | Over 66 Years | 614 (38%) |

**Table 2. Prevalence of common non-communicable diseases among the group of patients.**

| Disease | Total number affected (n = 1600) | Gender | | Age | | | | Percentage prevalence (95% CI) (n = 1600) |
|---|---|---|---|---|---|---|---|---|
| | | Male | Female | 20–35 Years | 36–50 Years | 51–65 Years | Over 66 Years | |
| Diabetes mellitus | 838 | 395 (47%) | 443 (53%) | 26 (3%) | 134 (16%) | 367 (44%) | 311 (37%) | 52.4% (49.9% - 54.8%) |
| Hypertension | 751 | 345 (46%) | 406 (54%) | 18 (2%) | 110 (15%) | 330 (44%) | 293 (39%) | 46.9% (44.5% - 49.4%) |
| Coronary heart disease (CHD) | 438 | 226 (52%) | 212 (48%) | 16 (4%) | 59 (13%) | 196 (45%) | 167 (38%) | 27.4% (25.2% - 29.6%) |
| Bronchial asthma | 167 | 77 (46%) | 90 (54%) | 02 (1%) | 21 (13%) | 77 (46%) | 67 (40%) | 10.4% (8.9% - 11.9%) |
| Arthritis | 125 | 55 (44%) | 70 (56%) | 04 (3%) | 12 (10%) | 53 (42%) | 56 (45%) | 7.8% (6.5%– 9.1%) |
| Hypo-hyperthyroidism | 122 | 59 (48%) | 63 (52%) | 04 (3%) | 22 (18%) | 52 (43%) | 44 (36%) | 7.6% (6.3%– 8.9%) |
| Peptic ulcer disease | 53 | 17 (32%) | 36 (68%) | 03 (6%) | 04 (8%) | 20 (38%) | 26 (49%) | 3.3% (2.4% - 4.2% |
| Mental disorders (e.g. depression, anxiety, dementia) | 51 | 26 (51%) | 25 (49%) | 02 (4%) | 06 (12%) | 21 (41%) | 22 (43%) | 3.2% (2.3%– 4.0%) |
| Heart failure | 50 | 27 (54%) | 23 (46%) | 00 (0%) | 08 (16%) | 18 (36%) | 24 (48%) | 3.1% (2.3%– 4.0%) |
| Chronic kidney disease (CKD) | 47 | 25 (53%) | 22 (47%) | 03 (6%) | 04 (9%) | 23 (49%) | 17 (36%) | 2.9% (2.1%– 3.8%) |
| Chronic obstructive airway disease | 40 | 19 (48%) | 21 (53%) | 00 (0%) | 09 (23%) | 15 (38%) | 16 (40%) | 2.5% (1.7%– 3.3%) |
| Stroke/ Transient ischaemic attack | 39 | 16 (41%) | 23 (59%) | 00 (0%) | 09 (23%) | 18 (46%) | 12 (31%) | 2.4% (1.7%– 3.2%) |
| Epilepsy | 30 | 12 (40%) | 18 (60%) | 01 (3%) | 02 (7%) | 17 (57%) | 10 (33%) | 1.9% (1.2%– 2.5%) |
| Atrial fibrillation | 18 | 07 (39%) | 11 (61%) | 00 (0%) | 02 (11%) | 09 (50%) | 07 (39%) | 1.1% (0.6%– 1.6%) |
| Chronic liver disease | 15 | 06 (40%) | 09 (60%) | 00 (0%) | 00 (0%) | 07 (47%) | 08 (53%) | 0.9% (0.5%– 1.4%) |
| Cancer | 13 | 04 (31%) | 09 (69%) | 00 (0%) | 02 (15%) | 08 (62%) | 03 (23%) | 0.8% (0.4%– 1.3%) |
| Interstitial lung disease | 2 | 02 (100%) | 00 (0%) | 00 (0%) | 00 (0%) | 01 (50%) | 01 (50%) | 0.13% (0%– 0.3%) |

association at a confidence level of 95%). Even solitary morbidities were found to be more prevalent among the females as represented in Fig 2.

The major NCDs that contributed to multimorbidity as illustrated in Fig 3 were diabetes mellitus, hypertension, followed by coronary heart disease. Smaller associations were also evident between diseases such as bronchial asthma and arthritis.

The comorbidities most commonly associated with each other were diabetes mellitus with hypertension (25%), followed by hypertension with coronary heart disease (12%) and diabetes mellitus with coronary heart disease (11%). Moreover, the combination of all three diseases diabetes mellitus with hypertension and coronary heart disease was also found to be comparatively common (7%) among the group of patients as represented in Table 3.

The mean number of disorders among patients with mental health diseases were 1.94 (± 0.95). The mean age of the patients with mental disorders was 61.31 years with a standard

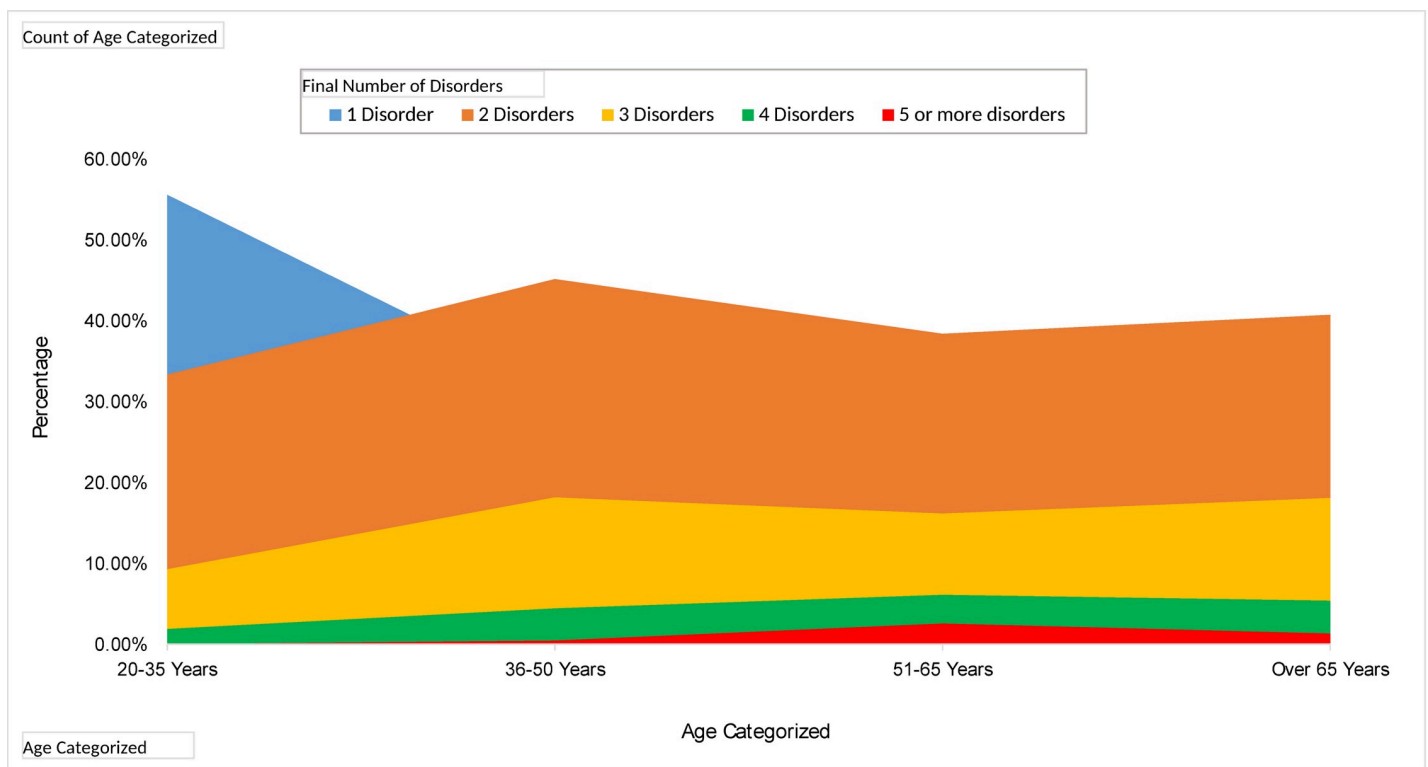

**Fig 1. Multimorbidity of diseases in different age groups.**

deviation of 12.16. The percentages of patients with mental disorders were nearly equal in relation to the male and female categories (51% vs 49%). Of all with mental disorders, 52.9% of the patients visited the tertiary health care facility, whereas 47.1% visited the primary care facility. Logistic regression analysis (Table 4) confirmed that as the number of disorders (multimorbidity) increases, the risk of developing any mental disorders increases by 7.25 (95% CI = 5.82–8.68). This was the only variable found to be significant through the regression analysis at 95% confidence level (p-value = 0.00 < 0.05).

## Discussion

This study gives valuable insights into the complexity of multimorbidity in Sri Lanka. To the best of our knowledge, the study is first of its kind to utilize a large sample to examine the burden, pattern and correlates multimorbidity among the adult population in Sri Lanka. This study is unique as it covers both primary and tertiary care settings.

It is known that multimorbidity is significantly associated with age. This finding was consistent across several studies, including those involving LMICs especially in India [8, 15]. In our study, the predominant age group was 51–65 years that accounted for 44% of the study population and closely followed by the older persons over the age of 65 years (38%). With advancing age, the number of comorbidities increases.

Multimorbidity is, however, not restricted to older patients. Being socioeconomically disadvantaged such as belonging to Lower middle-income countries (LMIC), speeds up the process of acquiring multimorbidity. Therefore, the population in countries such as Sri Lanka fall prey to multimorbidity earlier in life. In our study, 44% percent of patients aged 20–35 had two or more illnesses, which is alarming for a developing country like Sri Lanka. An Indian study also

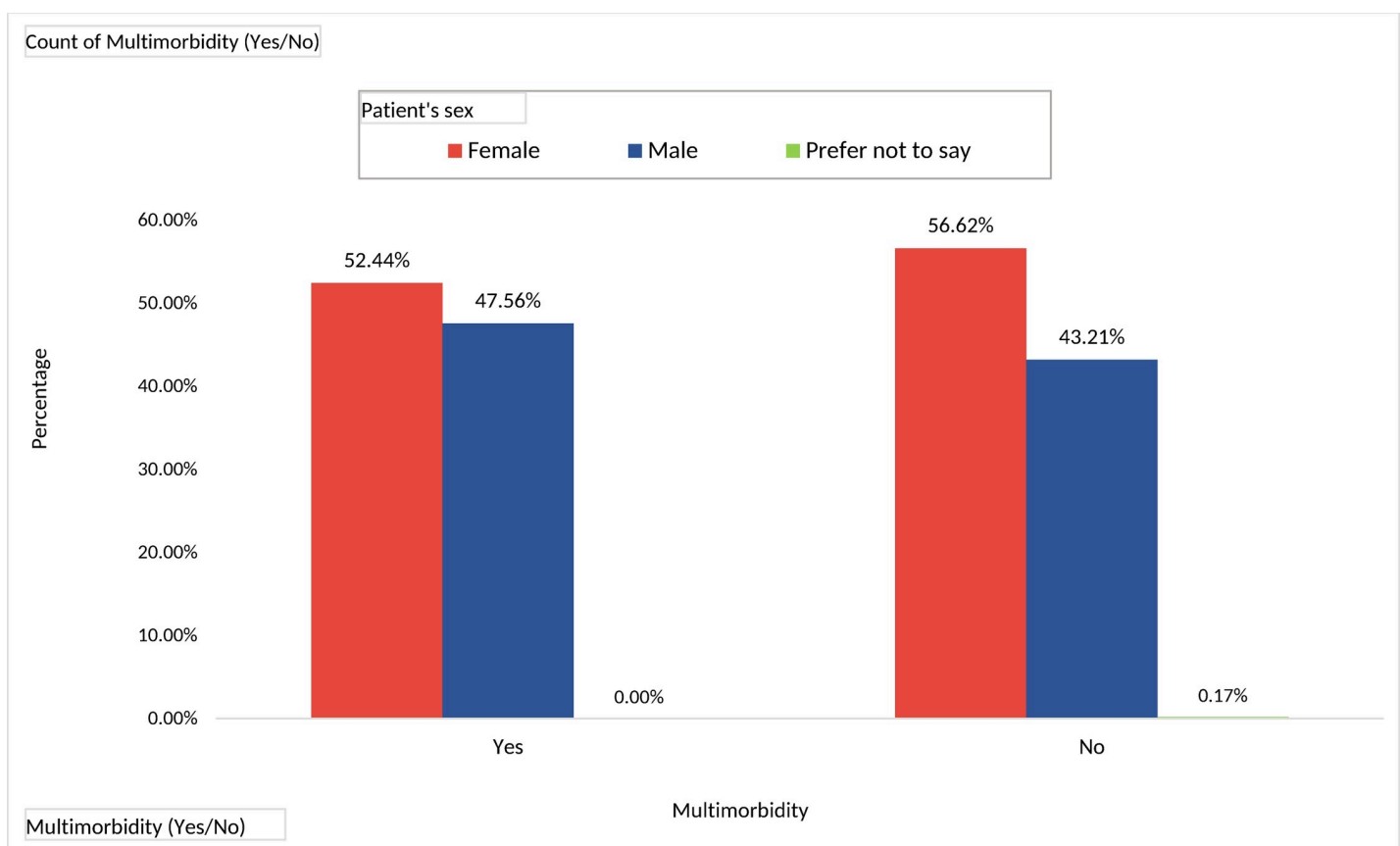

**Fig 2. Multimorbidity and its association with sex.**

shows that multimorbidity is prevalent among the young population as well [15]. In a study carried out in Australia (North West Adelaide Health Study), 4% of the 20–39 year age group, 15.0% of the 40–59 age group had multimorbidity [16]. Assuming multimorbidity is a problem of the aged undermines the real magnitude of the problem. This questionable assumption could have a serious economic impact for LMIC countries, including in healthcare resource allocation to subsets of the population. The rising burden of diseases as people age will pose a significant burden on an LMIC's development. Identifying that younger age groups are also affected should prompt to find active solutions for holistic care provision to this age group as they are not entitled by default to the care available to the geriatric population. This finding has important practical implications. Recently in Sri Lanka, several steps have been taken by policymakers to reduce risk factors that account for NCD's such as smoking, physical inactivity and unhealthy dietary patterns. The modification of these risk factors should begin at younger ages. At the policy level, screening for NCD's has been made mandatory for state-sector health care workers to get their promotions and increments. Similarly, modifications have been made in the school curricula to improve physical activity. An area of each town has been designated to host a walking, play or physical activity area to promote physical activity. All these measures will target younger individuals who belong to or constitute the future active workforce.

There is a female predominance in multimorbidity in worldwide studies even when gynae-cological diseases are not taken into account as morbidity. An extensive systematic review

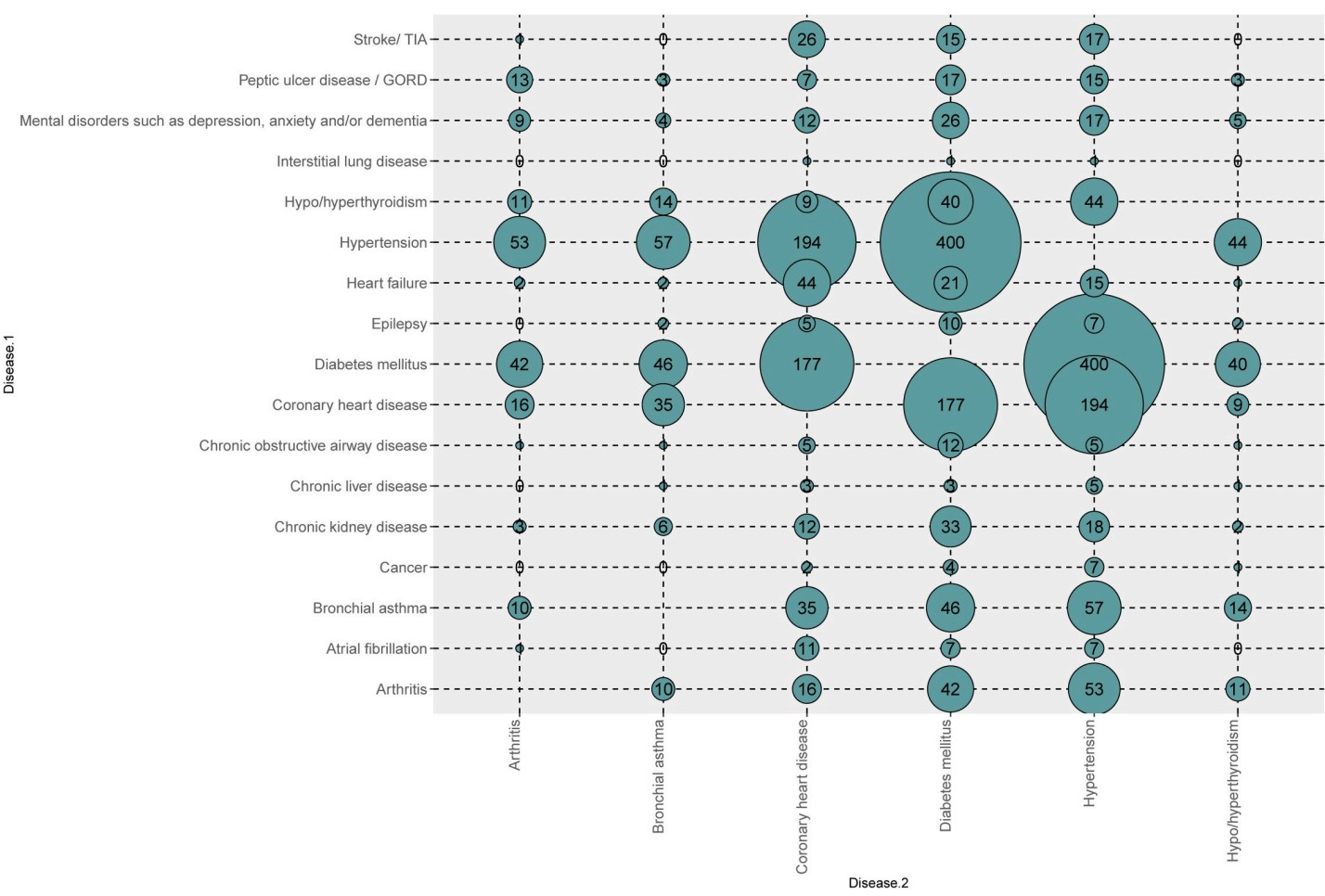

**Fig 3. Non-communicable diseases in multimorbidity.**

indicated that women had a greater prevalence of multimorbidity as compared to men [17]. There is growing evidence that women use more healthcare facilities, particularly government-funded free healthcare, compared to men. This could be one reason for the gender difference [17, 18]. Another reason could be due to the gender-based inequities in the health sector of Sri Lanka. The Sri Lankan health care system has been traditionally designed to support maternal

**Table 3. Prevalence of common multimorbidities among the group of patients.**

| Most Common Multimorbidities | Number affected (n = 1600) (%) | Gender | | Age | | | |
|---|---|---|---|---|---|---|---|
| | | Female (%) | Male (%) | 20–35 years (%) | 36–50 years (%) | 51–65 years (%) | Over 66 years (%) |
| Diabetes Mellitus & Hypertension | 400 (25%) | 218 (54.5%) | 182 (45.5%) | 08 (2%) | 66 (16.5%) | 178 (44.5%) | 148 (37%) |
| Hypertension & Coronary Heart Disease | 194 (12%) | 98 (50.5%) | 96 (49.5%) | 04 (2.1%) | 24 (12.4%) | 92 (47.4%) | 74 (38.1%) |
| Diabetes Mellitus & Coronary Heart Disease | 177 (11%) | 85 (48%) | 92 (52%) | 04 (2.3%) | 25 (14.1%) | 92 (52%) | 56 (31.6%) |
| Diabetes Mellitus, Hypertension & Coronary Heart Disease | 108 (7%) | 55 (50.9%) | 53 (49.1%) | 01 (0.9%) | 13 (12%) | 60 (55.6%) | 34 (31.5%) |

**Table 4. Logistic Regression analysis for mental health disorder by age, sex and number of physical disorders.**

| | Any mental health disorder | | |
| --- | --- | --- | --- |
| | Standardized Coefficients (95% CI) | Adjusted OR (95% CI) | p-value |
| Age (in years) | 0.21 (0.19, 0.24) | 1.00 (0.98, 1.03) | 0.80 |
| Male (Vs Female) | 0.34 (-0.25, 0.93) | 1.13 (0.63, 2.03) | 0.69 |
| Tertiary Care (Vs Primary Health Care) | 0.31 (-0.28, 0.91) | 1.12 (0.62, 2.02) | 0.71 |
| Multimorbidity (number of disorders) | 1.98 (0.9, 2.85) | 7.25 (5.82, 8.68) | 0.001 |

NB: The covariates used in the adjusted OR model are Age, Gender, Health Care Setting and the Number of health disorders (multimorbidity)

and child health outcomes over the years and Sri Lanka has achieved remarkable success in the field of maternal and child health care.

In our study, the commonest comorbidities found were diabetes and hypertension, of which diabetes was the commonest. Diabetes is a global public health concern and a common comorbidity in patients with hypertension [19, 20]. In urban South Asians context, diabetes and hypertension are most commonly encountered in other populations in the region [8]. The clustering of diabetes with hypertension and diabetes with coronary heart disease is also alarming. This clustering of disorders is well established in the past [21, 22]. Identifying clustering is essential as they signify underlying pathophysiology and risk factors to be similar, and the interventions to reduce the incidence of these illnesses also can take a similar approach. Our interventions could target individuals suffering from a given index disease such as diabetes who develop successive conditions such as CHD, stroke or CKD. Identifying and effective management of the initial condition could potentially lead to a lower incidence of the successive conditions.

In 2017, about 425 million people had diabetes worldwide, and approximately 80% lived in low- and middle-income countries [23]. The burden diabetes places on health care system are enormous due to its complications. The rising prevalence of diabetes in urban South Asia is attributed to the sedentary lifestyles and greater consumption of fast food rich in sugar and saturated fats that are supplemented by globalization [24, 25]. Having identified diabetes to be this common in an urban setting in Sri Lanka, necessary remedial steps for primary prevention can be implemented.

As Sri Lanka undergoes an epidemiological transition, mental health plays an integral part in morbidity and mortality of chronic diseases [26]. One of the most significant burdens of multimorbidity is its association with mental disorders. Our study confirmed that as the number of disorders increases, the risk of developing any mental disorders increases by 7.25 (95% CI = 5.82–8.68) and was the only variable that was significant in the regression analysis. Several studies in the past have also revealed that higher number of chronic conditions was associated with the poorer self-rated health, functional health measured using *activities of daily living* and *instrumental activities of daily living* and WHOQoL tools. This was reported in a study conducted across China, India, Russia, South Africa Mexico and Ghana [27]. In an Indian study, 66% of the older population was distressed physically, psychologically or both. Further, it was recognized that the number of morbidities was linked to poorer psychological wellbeing and increased disability [28]. A sizeable Scottish study published in the Lancet, also revealed that the prevalence of a mental disorder increased with the number of diseases [3]. Several other studies in Asia have also highlighted that rising multimorbidity affects mental health [29, 30].

Sri Lanka has a curative healthcare system, in which patients with specific diseases are cared for in selected specialized institutions (example: cancer care hospitals). This could have led to

the low proportion of patients with cancer or with any other specific diseases. In a meta-analysis on multimorbidity conducted in the South Asian setting it was revealed that none of the included studies was undertaken in a primary care setting [10]. Subsequently, a handful of studies had been done, inclusive of primary care [31]. This will be one of the few studies done in the primary care setting in South Asia assessing multimorbidity. As primary care plays an equally important role as tertiary care in health provision [10], these results will serve as an eye-opener.

In countries like Sri Lanka and other LMIC's, the dual burden of infectious disease and non-communicable diseases are pushing the health systems into peril. These countries have to continue to battle against infectious diseases while focusing on emerging multimorbidity [32]. Empowering of health systems to deal with multimorbidity requires training of healthcare workers to recognize the risk factors and offering health care advice to minimize the risks.

Our study shows that multimorbidity is indeed a problem in South Asia. Greater emphasis needs to be placed on further research into the area with the hope of providing better patient-centred care to those affected with multimorbidity. As the medical community is shifting into finer subspecialties a greater emphasis needs to be placed on treating patients as a whole as two or more chronic illness seems to cluster with an alarming rate in patients all over the globe. This change which requires horizontal integration needs to take place starting from medical school onwards and extend to the patient's bedside [21]. As Sri Lanka continues to battle the epidemic of NCDs, our current national NCD program should be tailor-made to care for patients with multiple morbidities. This will avoid fragmented care.

## Strengths and limitations

Being a descriptive study, ours has a few limitations that deserve mention. We relied on the details available in the clinic records without attempting to confirm the diagnosis by directly inquiring the patients' symptomatology. Hence, the accuracy of the diagnosis cannot be vouched entirely on. Certain diseases may have missed being diagnosed at all, and the patients who do not present to health care facilities for consultation have also been missed. Furthermore, a few more records were excluded owing to the lack of certain important demographic and disease-related information. Their exclusion could have biased our results to some extent. Third, the proportion of persons in our cohort with multimorbidity is based on the conditions we chose to define multimorbidity. We did not consider the severity of conditions or treatment given for conditions in this study. We also did not measure health outcomes, such as mortality and hospital admissions.

Although multimorbidity is very common in Sri Lanka and elsewhere in the world, it is not yet known how best to organise health services in order to manage these patients optimally. Details about the continuity of care and health service providers in the lifetime of the patients would have helped us to gain more insights into how health care should be organized to deliver optimal care. Information such as the number of visits to each health care provider, utilization of health services by patients would have provided us more details about health care service utilization that would have been useful in the interpretative exercise.

Our study also has several strengths. It provides for the first time in Sri Lanka, a detailed description of multimorbidity in both primary and tertiary care. The large sample size is also one important attribute. Besides, the large sample size allowed us to stratify patients into different age groups and also to see if certain diseases clustered. Identification of clustering is crucial as it has both policy and clinical implications. Provided the tertiary care study setting is one of the largest hospitals on the island, the results can be generalized to the country to a large extent. The inclusion of mental health conditions is also a vital feature of this study that constitutes a critical aspect of multimorbidity.

## Supporting information

**S1 Checklist. STROBE statement—Checklist of items that should be included in reports of cross-sectional studies.**
(DOCX)

**S1 Data.**
(XLSX)

**S1 File.**
(DOCX)

**S2 File.**
(DOCX)

## Acknowledgments

We wish to acknowledge the Departments of Medicine and Family Medicine of the Faculty of Medical Sciences, University of Sri Jayewardenepura, Sri Lanka for allowing to utilize the patient records for research purposes.

## Author Contributions

**Conceptualization:** Shamini Prathapan, Gunasekara Vidana Mestrige Chamath Fernando, Anne Thushara Matthias.

**Data curation:** Shamini Prathapan, Gunasekara Vidana Mestrige Chamath Fernando, Anne Thushara Matthias, Yashodara Bentota Mallawa Arachchige Charuni, Herath Mudiyanselage Gayan Abeygunawardhana, Batheegama Gamarachchige Gayasha Kavindi Somathilake.

**Formal analysis:** Shamini Prathapan, Gunasekara Vidana Mestrige Chamath Fernando, Anne Thushara Matthias, Yashodara Bentota Mallawa Arachchige Charuni, Herath Mudiyanselage Gayan Abeygunawardhana, Batheegama Gamarachchige Gayasha Kavindi Somathilake.

**Investigation:** Shamini Prathapan, Gunasekara Vidana Mestrige Chamath Fernando, Anne Thushara Matthias, Yashodara Bentota Mallawa Arachchige Charuni, Herath Mudiyanselage Gayan Abeygunawardhana, Batheegama Gamarachchige Gayasha Kavindi Somathilake.

**Methodology:** Shamini Prathapan, Gunasekara Vidana Mestrige Chamath Fernando, Anne Thushara Matthias.

**Project administration:** Shamini Prathapan, Gunasekara Vidana Mestrige Chamath Fernando, Anne Thushara Matthias, Yashodara Bentota Mallawa Arachchige Charuni, Herath Mudiyanselage Gayan Abeygunawardhana, Batheegama Gamarachchige Gayasha Kavindi Somathilake.

**Resources:** Shamini Prathapan, Gunasekara Vidana Mestrige Chamath Fernando, Anne Thushara Matthias, Yashodara Bentota Mallawa Arachchige Charuni, Herath Mudiyanselage Gayan Abeygunawardhana, Batheegama Gamarachchige Gayasha Kavindi Somathilake.

**Software:** Shamini Prathapan, Gunasekara Vidana Mestrige Chamath Fernando, Anne Thushara Matthias, Yashodara Bentota Mallawa Arachchige Charuni, Herath Mudiyanselage Gayan Abeygunawardhana, Batheegama Gamarachchige Gayasha Kavindi Somathilake.

**Supervision:** Shamini Prathapan, Gunasekara Vidana Mestrige Chamath Fernando, Anne Thushara Matthias.

**Validation:** Shamini Prathapan, Gunasekara Vidana Mestrige Chamath Fernando, Anne Thushara Matthias, Yashodara Bentota Mallawa Arachchige Charuni, Herath Mudiyanselage Gayan Abeygunawardhana, Batheegama Gamarachchige Gayasha Kavindi Somathilake.

**Visualization:** Shamini Prathapan, Gunasekara Vidana Mestrige Chamath Fernando, Anne Thushara Matthias, Yashodara Bentota Mallawa Arachchige Charuni, Herath Mudiyanselage Gayan Abeygunawardhana, Batheegama Gamarachchige Gayasha Kavindi Somathilake.

**Writing – original draft:** Shamini Prathapan, Gunasekara Vidana Mestrige Chamath Fernando, Anne Thushara Matthias, Yashodara Bentota Mallawa Arachchige Charuni, Herath Mudiyanselage Gayan Abeygunawardhana, Batheegama Gamarachchige Gayasha Kavindi Somathilake.

**Writing – review & editing:** Shamini Prathapan, Gunasekara Vidana Mestrige Chamath Fernando, Anne Thushara Matthias, Yashodara Bentota Mallawa Arachchige Charuni, Herath Mudiyanselage Gayan Abeygunawardhana, Batheegama Gamarachchige Gayasha Kavindi Somathilake.

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
