## [Decision Letter · Decision Letter 0]

23 Jul 2020

PONE-D-20-13239

The rising complexity of Multimorbidity in a middle income country

PLOS ONE

Dear Dr. Fernando,

Thank you for submitting your manuscript to PLOS ONE. After careful consideration, we feel that it has merit but does not fully meet PLOS ONE’s publication criteria as it currently stands. Therefore, we invite you to submit a revised version of the manuscript that addresses the points raised during the review process.

We look forward to receiving your revised manuscript.

Kind regards,

Alana T Brennan

Academic Editor

PLOS ONE

Journal Requirements:

2. Please ensure you have thoroughly discussed any potential limitations of this study within the Discussion section, including the impact of potentially confounding factors.

4. We note you have included a table to which you do not refer in the text of your manuscript. Please ensure that you refer to Table 1 in your text; if accepted, production will need this reference to link the reader to the Table.

5. Please ensure that you refer to Figure 3 in your text as, if accepted, production will need this reference to link the reader to the figure.

Reviewers' comments:

Reviewer's Responses to Questions

**Comments to the Author**

1. Is the manuscript technically sound, and do the data support the conclusions?

Reviewer #1: Partly

Reviewer #2: Partly

2. Has the statistical analysis been performed appropriately and rigorously? 

Reviewer #1: Yes

Reviewer #2: I Don't Know

3. Have the authors made all data underlying the findings in their manuscript fully available?

Reviewer #1: Yes

Reviewer #2: Yes

4. Is the manuscript presented in an intelligible fashion and written in standard English?

Reviewer #1: No

Reviewer #2: No

5. Review Comments to the Author

Reviewer #1: This article describes multi-morbidity in Sri Lanka based on data from a primary care center and a tertiary hospital. The article aims at adding to the literature by revising the characteristics of multi-NCD morbidity in Sri Lanka. The authors also look at the predictors of mental health in the selected population. The manuscript could be improved as suggested below.

MAJOR ISSUES

Overall.

1. Needs language revision – needs significant improvement on English

2. Not a clear research question/hypothesis. For instance, the stated goal is to describe multi morbidity however data is presented on association between multi-morbidity and mental health outcomes. It is not clear if mental health conditions is the outcome variable, if so, the hypothesis should be clearly stated.

Introduction

o The goal of the study is not clear. Is the goal to describe the characteristics of people diagnosed with NCD (i.e., age and sex [gender]?) or to describe the characteristics of people diagnosed with 2 or more NCDs?

Methods

Sample size: Description of sample size estimation is not clear

o If the study is descriptive in nature, what is the significance level for? Is this an analytical or descriptive study?

Needs to provide rationale for not including all patients in the registry and limiting it to a sample size calculation.

Were the people excluded different than those that remained in the study?

MINOR ISSUES

Overall

•

• Data was collected only for individuals over 20 years of age but no rationale is provided for this selection.

Abstract.

• Conclusions not supported by data

• Abstract needs to be rewritten

• Reorder the introduction

• Not clear why only regression for mental health.

Figures and tables.

• Figure 3 is not high resolution and not clear

• No need for tables under figures 1 and 2

Introduction

o Might need some reformatting of paragraphs

• Methods

o Selection of clinic/hospital needs to be better justified (is it representative of the population, why? If not, how to avoid bias)

o Again, provide rationale for selecting only individuals over 20 years.

o Needs to add rationale behind selecting diagnosis for inclusion.

o Formatting of age groups is different (line 109)

o Provide rationale for age groups

Methods. A. Sample Size

o Is the data extraction form provided?

Data analysis

Why restricting to age 18 and older when the eligibility states age 20 and older?

o Suggest to use either SPSS or R. Indicate version, year, etc. full software citation.

• Results

o Not clear what is meant by “by the time they reach half the decade” – it would be best to describe the age group.

o The use of “increase” in line 148 suggests a baseline. I suggest to consider use another term that more accurately describes the comparision being made.

o Use n= XXX as opposed to XX/1600

o How does the proportion of multimorbidity relates to original size of age groups (an adjusted rate might be useful)?

• Discussion

o Could be more focused.

o Needs to better discuss the strengths and limitations of the study.

Reviewer #2: "The rising complexity of Multimorbidity in a middle income country " emphasizes important health issues. Below are some suggestions to strengthen the manuscript

Comments

1. Methods section: Please add subsections and reorganize as necessary to improve clarity and readability

2. Methods (line 109) please justify why these age categories were used, if possible referencing previous studies for compatibility

3. Methods (line 128) states the analysis was restricted " to those aged 18 years and older", however line 101 states "patients over 20 years ...". Please correct or clarify.

4. Table 2 Please clarify in the footnote which covariates were uncluded in the adjusted OR model. Would it be possible to present unadjusted and adjusted. Could additional factors be included in the adjusted model?

5. It is clear ethical approval was obtained from the relevant institute. Patient data was anonymous. Did patients give permission to the institute to use their data?

6. Please carefully check English language usage throughout the text.

6. PLOS authors have the option to publish the peer review history of their article (what does this mean?). If published, this will include your full peer review and any attached files.

Reviewer #1: No

Reviewer #2: No

---

## [Author Response · Author response to Decision Letter 0]

8 Sep 2020

In the attached response to the reviewers file.

---

## [Decision Letter · Decision Letter 1]

8 Oct 2020

PONE-D-20-13239R1

The rising complexity of Multimorbidity in a middle income country

PLOS ONE

Dear Dr. Fernando,

Thank you for submitting your manuscript to PLOS ONE. After careful consideration, we feel that it has merit but does not fully meet PLOS ONE’s publication criteria as it currently stands. Therefore, we invite you to submit a revised version of the manuscript that addresses the points raised during the review process.

ACADEMIC EDITOR:

The reviewers and I both noted that you have done a nice job addressing the previous comments. There are some additional suggestions, in particular those raised by Reviewer 3, that would greatly strengthen the manuscript, if addressed. Please note, too, the suggestions for modifying the title and changing the language describing the mental health findings to better reflect the work you have done.

We look forward to receiving your revised manuscript.

Kind regards,

Andrea Gruneir

Academic Editor

PLOS ONE

Reviewers' comments:

Reviewer's Responses to Questions

**Comments to the Author**

1. If the authors have adequately addressed your comments raised in a previous round of review and you feel that this manuscript is now acceptable for publication, you may indicate that here to bypass the “Comments to the Author” section, enter your conflict of interest statement in the “Confidential to Editor” section, and submit your "Accept" recommendation.

Reviewer #2: (No Response)

Reviewer #3: (No Response)

2. Is the manuscript technically sound, and do the data support the conclusions?

Reviewer #2: Yes

Reviewer #3: Partly

3. Has the statistical analysis been performed appropriately and rigorously? 

Reviewer #2: Yes

Reviewer #3: I Don't Know

4. Have the authors made all data underlying the findings in their manuscript fully available?

Reviewer #2: Yes

Reviewer #3: Yes

5. Is the manuscript presented in an intelligible fashion and written in standard English?

Reviewer #2: Yes

Reviewer #3: Yes

6. Review Comments to the Author

Reviewer #2: Thank you for addressing almost all comments. Please additionally add subsection headers within the methods section

Reviewer #3: Examples of literature on MM in Intro and Discussion are largely from high income countries - when there are a reasonable number of publications from lower income countries that might be more appropriate.

Table 1 is useful, but could be more useful to have info by age and sex, plus a similar table of most common multimorbidities. Would want table of patient sociodemographic characteristics. The authors could please specify which variables were included in the adjusted regression analyses?

More details about the regression analyses are needed - describing the stepped process by which you did both sets of regression analyses - and especially the analysis including mental health variables.

And the conclusions about mental health need to be tempered - this is a cross-sectional study "...as the number of disorders increases, the risk of developing any mental disorders increases..."

The term "elderly" is considered pejorative, please instead use older persons, older adult, older population.

One of the biggest findings here is the high prevalence in 'young" age groups. This could be part of another analysis - (narrower age groups (20-29, 30-39, etc) and look at age of diagnosis - but for this publication, authors could emphasize this interesting finding about high MM prevalence in young age groups.

Finally, the title is misleading - it describes the burden of multimorbidity in a sample of patients, not its complexities.

7. PLOS authors have the option to publish the peer review history of their article (what does this mean?). If published, this will include your full peer review and any attached files.

Reviewer #2: No

Reviewer #3: No

---

## [Author Response · Author response to Decision Letter 1]

5 Nov 2020

In the attached document entitled "Response to the reviewers"

---

## [Editor Report · Decision Letter 2]

13 Nov 2020

PONE-D-20-13239R2

The rising complexity and burden of multimorbidity in a middle-income country

PLOS ONE

Dear Dr. Fernando,

Thank you for submitting your manuscript to PLOS ONE. After careful consideration, we feel that it has merit but does not fully meet PLOS ONE’s publication criteria as it currently stands. Therefore, we invite you to submit a revised version of the manuscript that addresses the points raised during the review process.

ACADEMIC EDITOR:

Thank you for taking the time to revise and resubmit your manuscript. I think that you have done a nice job of addressing the Reviewers' comments throughout the text. I would just like to see some modifications to the tables so that they are easier to read. In all tables, it would be helpful to have the sample size shown within the column headers so that the denominator for each cell is clear. For Table 2, you should consider having a total column (where you can show the prevalence of each condition for the full sample) and then the prevalence stratified by each sex and age; you sort of have this but you need the denominator for each column and perhaps a bit more cleaning up. You may want to consider similar adjustments to Table 3. There is a fair bit of data in your study and anything you can do to help your readers understand it would be useful.

We look forward to receiving your revised manuscript.

Kind regards,

Andrea Gruneir

Academic Editor

PLOS ONE

---

## [Author Response · Author response to Decision Letter 2]

15 Nov 2020

Please see in the "Response to Reviewers" document.

---

## [Editor Report · Decision Letter 3]

25 Nov 2020

The rising complexity and burden of multimorbidity in a middle-income country

PONE-D-20-13239R3

Dear Dr. Fernando,

We’re pleased to inform you that your manuscript has been judged scientifically suitable for publication and will be formally accepted for publication once it meets all outstanding technical requirements.

Kind regards,

Andrea Gruneir

Academic Editor

PLOS ONE

Additional Editor Comments (optional): Thank you for your patience and for continuing to make revisions to this manuscript. I think that there are still some modifications to your tables that are required to make them as readable as possible, but also that there is no reason to hold back your manuscript at this point. Congratulations on this work!
---

## [Editor Report · Acceptance letter]

2 Dec 2020

PONE-D-20-13239R3 

The rising complexity and burden of multimorbidity in a middle-income country 

Dear Dr. Fernando:

I'm pleased to inform you that your manuscript has been deemed suitable for publication in PLOS ONE. Congratulations! Your manuscript is now with our production department. 

Kind regards, 

on behalf of

Dr. Andrea Gruneir 

Academic Editor

PLOS ONE